# The Relationships between Dopaminergic, Glutamatergic, and Cognitive Functioning in 22q11.2 Deletion Syndrome: A Cross-Sectional, Multimodal ^1^H-MRS and ^18^F-Fallypride PET Study

**DOI:** 10.3390/genes13091672

**Published:** 2022-09-19

**Authors:** Carmen F. M. van Hooijdonk, Desmond H. Y. Tse, Julia Roosenschoon, Jenny Ceccarini, Jan Booij, Therese A. M. J. van Amelsvoort, Claudia Vingerhoets

**Affiliations:** 1Department of Psychiatry and Neuropsychology, School for Mental Health and Neuroscience (MHeNs), University of Maastricht, 6226 NB Maastricht, The Netherlands; 2Rivierduinen, Institute for Mental Health Care, 2333 ZZ Leiden, The Netherlands; 3Department of Circulation and Medical Imaging, Norwegian University of Science and Technology, NO-7491 Trondheim, Norway; 4Department of Nuclear Medicine and Molecular Imaging, Division of Imaging and Pathology, KU Leuven, B-3000 Leuven, Belgium; 5Department of Radiology and Nuclear Medicine, Amsterdam UMC, University of Amsterdam, 1105 AZ Amsterdam, The Netherlands

**Keywords:** 22q11.2 deletion syndrome, dopamine, glutamate, psychotic disorders, cognitive dysfunction

## Abstract

Background: Individuals with 22q11.2 deletion syndrome (22q11DS) are at increased risk of developing psychosis and cognitive impairments, which may be related to dopaminergic and glutamatergic abnormalities. Therefore, in this exploratory study, we examined the association between dopaminergic and glutamatergic functioning in 22q11DS. Additionally, the associations between glutamatergic functioning and brain volumes in 22q11DS and healthy controls (HC), as well as those between dopaminergic and cognitive functioning in 22q11DS, were also examined. Methods: In this cross-sectional, multimodal imaging study, glutamate, glutamine, and their combined concentration (Glx) were assessed in the anterior cingulate cortex (ACC) and striatum in 17 22q11DS patients and 20 HC using 7T proton magnetic resonance spectroscopy. Ten 22q11DS patients also underwent ^18^F-fallypride positron emission tomography to measure dopamine D_2/3_ receptor (D_2/3_R) availability in the ACC and striatum. Cognitive performance was assessed with the Cambridge Neuropsychological Test Automated Battery. Results: No significant associations were found between ACC or striatal (1) glutamate, glutamine, or Glx concentrations and (2) D_2/3_R availability. In HC but not in 22q11DS patients, we found a significant relationship between ACC volume and ACC glutamate, glutamine, and Glx concentration. In addition, some aspects of cognitive functioning were significantly associated with D_2/3_R availability in 22q11DS. However, none of the associations remained significant after Bonferroni correction. Conclusions: Although our results did not reach statistical significance, our findings suggest an association between glutamatergic functioning and brain volume in HC but not in 22q11DS. Additionally, D_2/3_R availability seems to be related to cognitive functioning in 22q11DS. Studies in larger samples are needed to further elucidate our findings.

## 1. Introduction

22q11.2 deletion syndrome (22q11DS), with a prevalence of 1 in 2000–4000 births, is a relatively common genetic disorder that is characterized by a microdeletion on chromosome 22 locus q11.2 [1]. The typically deleted region contains approximately 90 genes [2]. Half of these are protein-coding genes, most of which are expressed in the brain [2]. The phenotypic expression of 22q11DS is highly heterogeneous and includes, among others, palatal anomalies, hypocalcemia, and congenital heart diseases [3]. Furthermore, the lifetime risk of developing a psychotic disorder for individuals with 22q11DS is 20–40% [4], compared to 1–3% in the general population [5]. Individuals with 22q11DS often experience cognitive impairments, which can decline further with age [6]. Moreover, the cognitive decline is steeper in individuals with 22q11DS who develop a psychotic disorder [6,7]. 

Two of the genes within the deleted region in 22q11DS are the catechol-O-methyltransferase (*COMT*) and proline dehydrogenase (*PRODH*) genes. The *COMT* gene encodes the *COMT* enzyme, which catabolizes extracellular dopamine. Dopamine levels in frontal brain regions are especially thought to be affected by the haploinsufficiency of the *COMT* gene [8]. Previous imaging studies have investigated dopaminergic functioning in subjects with 22q11DS and reported increased striatal dopamine synthesis capacity [9], as well as reduced dopamine D_2/3_ receptor (D_2/3_R) binding in frontal brain areas of individuals with 22q11DS compared to healthy controls [10].

The *PRODH* gene encodes the *PRODH* enzyme, which plays a role in the degradation of proline. The degradation of proline generates glutamate. Proline and glutamate can both, among other functions, activate the glutamatergic N-methyl-D-aspartate (NMDA) receptor [11,12]. It has been hypothesized that reduced *PRODH* enzyme activity in 22q11DS due to haploinsufficiency of the *PRODH* gene results in elevated proline levels [13]. Hyperprolinemia is a common finding in patients with 22q11DS [13,14,15]. Elevated proline levels may cause elevated activation of the NMDA receptor and excessive glutamate release [11,16]. Excessive glutamate levels are neurotoxic and can lead to neuronal injury and subsequent cell death [17]. Patients with excitotoxic damage are expected to have worse outcomes (i.e., more neurodegeneration, cognitive deficits, and negative symptoms) than patients without excitotoxic damage [18]. Due to *PRODH* haploinsufficiency, glutamate neuroexcitotoxicity may occur more frequently in 22q11DS relative to healthy individuals, which might explain the reduced cortical brain volumes reported in these patients [19]. Nevertheless, recent studies did not reveal significant alterations in glutamatergic functioning, as assessed by proton magnetic resonance spectroscopy (^1^H-MRS), in the ACC or the striatum of patients with 22q11DS compared to healthy controls [20,21]. However, increased hippocampal glutamate and Glx (glutamate and glutamine combined) concentrations were found in 22q11DS patients who developed schizophrenia compared to 22q11DS patients who did not [22]. 

In schizophrenia and corresponding at-risk populations, increased striatal dopamine synthesis capacity has been a well-replicated finding [23,24,25,26]. However, in recent years, additional theories have been posited, suggesting that disrupted cortical glutamatergic functioning might underlie these striatal dopaminergic alterations in schizophrenia [27]. Preclinical studies, as well as in vivo studies, have demonstrated a relationship between dopaminergic and glutamatergic functioning. For example, the administration of ketamine, which blocks the NMDA receptors on y-aminobutyric acid GABAergic interneurons, resulting in the disinhibition of glutamatergic neurons and increased striatal dopamine levels in rodents [28]. Furthermore, positron emission tomography (PET) studies showed that the administration of ketamine increased synaptic dopamine levels in the striatum of healthy human volunteers [29,30]. Finally, a multimodal ^18^F-FDOPA PET and ^1^H-MRS imaging study reported an inverse relation between glutamate concentration in the ACC and striatal dopamine synthesis capacity in patients with psychosis [31]. 

In summary, possible alterations in dopaminergic and glutamatergic systems in individuals with 22q11DS might explain the increased risk of developing a psychotic disorder, as well as the increased prevalence of cognitive impairments in these patients. Although previous studies have examined dopaminergic [9,10,32] and glutamatergic functioning [20,21] in individuals with 22q11DS, to the best of our knowledge, no study has examined whether cortical and striatal glutamatergic and dopaminergic measures are correlated in individuals with 22q11DS. Therefore, we investigated glutamate, glutamine, and Glx concentrations in the ACC and striatum in relation to frontal and striatal dopamine D_2/3_R availability in individuals with 22q11DS using ^1^H-MRS and ^18^F-fallypride PET, respectively. Comparable to findings in patients with psychosis [31], we hypothesized that in 22q11DS, ACC glutamate concentration would be inversely correlated with striatal dopamine D_2/3_R availability. Additionally, we investigated the association between (1) glutamate, glutamine, and Glx concentrations in the ACC and striatum and (2) ACC volumes in individuals with 22q11DS and healthy volunteers. We hypothesized that higher frontal glutamate, glutamine, and Glx concentrations would be related to lower ACC volumes in patients. The third aim of the present study was to explore the association between cognitive functioning and dopamine D_2/3_R availability in the ACC and striatum in individuals with 22q11DS.

## 2. Materials and Methods

### 2.1. Participants

A total of 17 non-psychotic adult individuals with 22q11DS were recruited through the National Adult 22q11DS Outpatient Clinic at Maastricht University Medical Centre and through the Dutch 22q11DS family network. In addition, 20 age- and sex-matched healthy volunteers were enrolled via social media and advertisement. All participants were recruited as part of a 7T 1H-MRS study [21]. In addition, a subgroup of 22q11DS patients participated in an ^18^F-fallypride PET study [32]. Recruitment was carried out as previously described [21,32]. Briefly, inclusion criteria were (1) 18–65 years of age and, for adults with 22q11DS, (2) the mental capacity to give informed consent; and (3) a confirmed diagnosis of 22q11DS by fluorescence in situ hybridization (FISH), microarray, or multiplex ligation-dependent probe amplification (MLPA). For both groups, exclusion criteria were (1) a history of psychosis as determined by the Mini International Neuropsychiatric Interview (MINI [33], (2) recreational drug use 4 weeks before participation, (3) previous or current use of stimulant or antipsychotic medication, (4) contraindications for PET and/or magnetic resonance imaging (MRI), and for female participants, (5) pregnancy. Ethical permission was obtained from the Medical Ethical Committee of Maastricht University (The Netherlands; METC142046, NL49834.068.14). Written informed consent was obtained from every participant. 

### 2.2. Procedure and Instruments

All subjects underwent ^1^H-MRS to assess glutamate, glutamine, and Glx concentrations in the right striatum and ACC. Furthermore, a subgroup of ten individuals with 22q11DS underwent ^18^F-fallypride PET to assess dopamine D_2/3_R availability in the putamen, caudate nucleus (CNC), ventral striatum (VST), and ACC. Cognitive performance was assessed in all subjects with the Cambridge Neuropsychological Test Automated Battery (CANTAB) [34]. Seven cognitive domains were assessed with multiple tasks: visual learning and memory, verbal learning and memory, working memory, attention and vigilance, processing speed, reasoning and problem solving, and social cognition (see Appendix A). The FSIQ was determined by the use of the shortened version of the Wechsler Adult Intelligence Scale, version 3 (WAIS-III) [35]. The MINI was used to verify the absence of psychiatric disorders [33]. All tests were administered on the same day as the ^1^H-MRS scan. Additionally, urine drug screening was performed to assure all subjects were free of recreational drugs (i.e., amphetamines, benzodiazepines, cannabis, cocaine, methamphetamines, and opiates). Furthermore, all female participants tested negative for pregnancy in a separate urine screening.

### 2.3. ^1^H magnetic Resonance Spectroscopy and Structural MRI

^1^H-MRS spectra were acquired on a MAGNETOM 7T MR scanner (Siemens Healthineers, Erlangen, Germany) with a stimulated echo acquisition mode (STEAM) sequence (TE = 6.0 ms, TR = 5.0 s, NA = 64, flip angle = 90°) [36]. Spectroscopy voxels were manually placed on the right striatum and ACC (Figure 1). LCModel version 6.3-1L [37] was used to analyze the ^1^H-MRS spectra by use of a GAMMA-simulated basis set [38] and to estimate concentrations of glutamate, glutamine, and Glx. Metabolite analyses were restricted to spectra with a Cramer–Rao lower bound ≤ 20%. Glutamate, glutamine, and Glx concentrations were corrected for the proportion of CSF as described in [39]. An anatomical T_1_-weighted image was obtained using a magnetization-prepared two rapid acquisition gradient-echo (MP2RAGE) sequence (TR = 4.5 s, TE = 2.39 ms, TI_1_ = 0.90 s, TI_2_ = 2.75 s, flip angle_1_ = 5°, flip angle_2_ = 3°, voxel size = 0.9 mm isotropic, matrix size = 256 × 256 × 192) [40]. ACC volumes were calculated by use of Freesurfer, version 6 [41], as described in [42]. A detailed description of the ^1^H-MRS procedure can be found in [21].

### 2.4. Positron Emission Tomography

Before the start of the PET scan, a 10 min low-dose ^68^Ge/^68^Ga transmission scan was obtained for attenuation correction purposes. Subsequently, approximately 200 MBq ^18^F-fallypride was administered, followed by 120 min of dynamic PET acquisition, as described in [43]. The previously collected T_1_-weighted image was used for coregistration purposes. SPM2 (Wellcome Trust, UK) was used to realign the ^18^F-fallypride frames. The PMOD software package (v. 3.6, PMOD Technologies Ltd., Zurich, Switzerland) was used to execute an automatic preprocessing protocol. Realigned PET images were coregistered to the individual T_1_-weighted image. Afterwards, the individual T_1_-weighted images were spatially normalized to standard Montreal Neurological Institute (MNI) space in PMOD. PET images were spatially normalized using the same spatial transformation. For each patient, the T_1_-weighted images were segmented into white matter, grey matter, and cerebrospinal fluid within native MRI space. The PMOD PNEURO tool was used for automatic delineation of the regions of interest (ROIs) by use of the N30R83 Hammers probabilistic atlas [44]. The atlas was adjusted to the T_1_-weighted scan of the subject. The following ROIs were investigated: (1) ACC, mean, left, and right; (2) putamen; (3) CNC; (4) VST; and (5) cerebellum (i.e., cerebellar hemispheres without the vermis; reference region) [44]. Subsequently, the linear extension of the SRTM (LSRRM) [45] was used to estimate kinetic parameters and the time–activity curves (TACs) for all striatal and frontal ROIs. Using an in-house MATLAB (version 6.5) script, ^18^F-fallypride binding potential (BP_ND_) was estimated in each ROI [45]. A detailed description of the PET procedure can be found in [32,43].

### 2.5. Statistical Analyses

All statistical analyses were performed in IBM SPSS Statistics (version 22). Differences in sample characteristics, including age, sex, selective serotonin reuptake inhibitor (SSRI) use, and FSIQ, were assessed using chi-square, Fisher’s exact, or Mann–Whitney U tests. Subsequently, cognitive domain scores were calculated by (1) reverse coding the scales of some outcome measures such that higher scores corresponded to better performance on all tasks, (2) calculating z-scores and removing outliers (i.e., z-scores lower than −3 or higher than 3), and (3) summing all z-scores within a cognitive domain and dividing by the number of outcome measures within the domain. A composite score was calculated by computing the sum of all seven domain scores. Finally, given the limited sample size and its robustness to the influence of outliers, Spearman’s correlation coefficient was used to examine the associations between (1) striatal and frontal dopaminergic and glutamatergic functioning in individuals with 22q11DS, (2) striatal and ACC glutamatergic functioning and ACC volumes in individuals with 22q11DS and healthy controls, and (3) striatal and frontal dopaminergic and cognitive functioning in 22q11DS. Bonferroni correction was used to correct for multiple testing. Consequently, for the first, second, and third objectives, *p*-values < 0.0083 (0.05/(3 [^1^H-MRS metabolites] × 2 [^1^H-MRS brain regions]), <0.0125 (0.05/4 [ACC volumes]), and <0.00555 (0.05/9 [7 cognitive domains, composite score, and FSIQ]) were considered significant, respectively.

## 3. Results

### 3.1. Demographics

Demographic details of participants (i.e., 22q11DS and healthy controls that underwent MRI scanning, as well as a subgroup of patients with 22q11DS that also underwent PET scanning) are shown in Table 1. There were no between-group differences in sex, age, smoking status, and SSRI use between 22q11DS individuals and healthy controls who underwent MRI. However, as expected, patients with 22q11DS had significantly lower FSIQ-scores compared to healthy controls (U = 4.50, *p* < 0.001).

### 3.2. Association between Dopaminergic and Glutamatergic Functioning in 22q11DS

Within the 22q11DS group, glutamate, glutamine, and Glx concentrations in the ACC or striatum were not significantly correlated with mean dopamine D_2/3_R availability in the ACC, CNC, putamen, or VST (Table 2). In addition, no significant associations were found between glutamate, glutamine, or Glx concentrations in the ACC or striatum and left or right D_2/3_R availability in the CNC, putamen, or VST (Appendix A).

### 3.3. Association between Glutamatergic Functioning and ACC Volumes in 22q11DS and Healthy Controls

Within the 22q11DS group, no significant correlations were found between left and right rostral and caudal ACC volumes and glutamate, glutamine, or Glx concentrations in the ACC or striatum (Table 3). Furthermore, within the healthy control group, significant positive associations were found between right rostral ACC volume and glutamate concentration in the ACC (effect size measure, *r* = 0.49), left caudal ACC volume, and glutamine concentration in the ACC (effect size measure *r* = 0.51), as well as between right caudal ACC volume and Glx concentration in the ACC (effect size measure *r* = −0.53). However, these associations were no longer significant after Bonferroni correction.

### 3.4. Association between Cognitive Functioning and Dopamine D_2/3_ Receptor Availability in 22q11DS

One 22q11DS subject was excluded from the analyses that focused on the cognitive domain attention due to an extreme value. There were no outliers for the other cognitive domains, composite score, or FSIQ. Within the 22q11DS group, mean, left, and right dopamine D_2/3_R availability in the CNC, putamen, and VST were not significantly related to any of the seven cognitive domains, the composite score, or FSIQ (Table 4 and Appendix A), except for dopamine D_2/3_R availability in the left VST and verbal memory (effect size measure, *r* = −0.70). However, after Bonferroni correction, this association did not remain significant. Furthermore, visual memory, executive functioning, and the composite score were significantly correlated with dopamine D_2/3_R availability in the ACC (although not significant after Bonferroni correction). The results remained the same after correcting for ACC volume (i.e., left, right, caudal, and rostral ACC volumes combined). The association between cognitive and glutamatergic functioning was previously reported in the same sample and is therefore not re-examined in this study [21].

## 4. Discussion

The aims of this study were threefold: (I) to investigate the association between dopaminergic and glutamatergic markers in 22q11DS, (II) to examine the association between glutamatergic functioning and ACC volumes in 22q11DS and healthy controls, and (III) to investigate the association between cognitive functioning and dopamine D_2/3_R availability in 22q11DS. Although we did not find significant associations after Bonferroni correction between any of the abovementioned outcomes, our results provide useful insights. Despite the limited sample size, some associations reached statistical significance with medium-to-large effect sizes. 

### 4.1. Association between Dopaminergic and Glutamatergic Functioning in 22q11DS

We did not find a significant association between dopaminergic and glutamatergic functioning in 22q11DS. This result is not in line with previous findings in patients with psychosis [31] and individuals at ultra-high risk of psychosis [46]. The lack of associations between dopaminergic and glutamatergic markers in our study is likely related to the small sample size, as only ten participants underwent both dopaminergic and glutamatergic imaging. Another speculative explanation is that the participants in our study did not have pronounced psychotic symptoms, as opposed to the participants in [31,46], which employed ^18^F-DOPA PET to investigate dopamine synthesis capacity. This could suggest that the association between glutamatergic and dopaminergic functioning might be a state characteristic for psychotic symptoms. However, this is speculative and should be examined in future research. Despite the lack of statistical significance, we did report some medium effect sizes comparable to the effect size reported in [31]. Therefore, we cannot rule out that a significant association between glutamatergic and dopaminergic markers exists in 22q11DS. Moreover, dopamine D_2/3_R availability in the striatum, as measured with PET or single-photon emission computed tomography (SPECT), is determined by multiple aspects: endogenous concentrations of dopamine in the synaptic cleft, affinity of the used radiotracer for the dopamine D_2/3_R, and receptor density [47,48]. Therefore, compensatory mechanisms that cancel each other out may explain the absence of associations between dopamine D_2/3_R availability and ACC glutamate/glutamine/Glx concentrations in 22q11DS. Finally, Jauhar et al. [31] did not find a significant relation between Glx concentration in the ACC and striatal dopamine synthesis capacity in patients with psychosis, which is in line with our findings. Future studies should be conducted with a larger sample, making use of multimodal imaging techniques to further elaborate these exploratory findings and to advance our understanding in this area. 

### 4.2. Association between Glutamatergic Functioning and ACC Volumes in 22q11DS and Healthy Controls

Prior to Bonferroni correction, we found an association between right rostral ACC volume and glutamate concentration in the ACC, between left caudal ACC volume and glutamine concentration in the ACC, as well as between right caudal ACC volume and Glx concentration in the ACC in healthy controls. However, no such associations were found in 22q11DS. This suggests that the associations between ACC volumes and glutamate/glutamine/Glx concentrations in the ACC may differ between groups. However, additional research is needed to elucidate this phenomenon. Schizophrenia and 22q11DS are characterized by a loss of brain volume [19,49], and previous research has suggested that the glutamatergic system might be involved in the mechanism underlying this loss of brain volume [50,51]. The glutamatergic system is of particular interest due to its potential to cause neuroexcitotoxicity, which may lead to reduced grey matter volume. The excitotoxicity hypothesis of schizophrenia proposes that in at least a subgroup of patients with schizophrenia, excitotoxic neuronal cell death occurs in cortical and hippocampal regions via the disinhibition of glutamatergic projections to these regions [18]. Multiple studies have reported associations between glutamatergic and structural measures in patients with psychosis. In unmedicated patients with schizophrenia but not healthy controls, increased glutamatergic levels in the hippocampus have been associated with reduced hippocampal volume [52]. In addition, Plitman et al. [53] found a negative association between Glx levels in the precommissural dorsal caudate and precommissural caudate volume in patients with a first non-affective episode of psychosis. This was not the case for healthy controls. Our preliminary results are in line with these findings, suggesting that the association between glutamatergic functioning and brain volume differs between patients with psychosis and controls. Because 22q11DS is associated with an increased risk of developing psychosis [4], neuroexcitotoxicity due to excessive glutamate might also occur more frequently in at least a subgroup of individuals with 22q11DS who develop psychosis. A previous study did not reveal increased hippocampal glutamate, glutamine, or Glx levels in non-psychotic 22q11DS patients compared to controls but revealed increased hippocampal glutamate and Glx concentrations in 22q11DS patients who developed schizophrenia compared to 22q11DS patients who did not [22]. This suggests that patients who develop psychosis might benefit from drugs that affect the glutamatergic system. Further studies should be conducted to elaborate on this hypothesis.

### 4.3. Association between Cognitive Functioning and Dopamine D_2/3_ Receptor Availability in 22q11DS

Within the 22q11DS group, the association between dopamine D_2/3_R availability in the left VST and verbal memory, as well as the associations between dopamine D_2/3_R availability in the ACC and visual memory, executive functioning, and the composite score, reached statistical significance. Again, the effect sizes are noteworthy (i.e., corresponding to strong effects [54]). This suggests that our hypothesis of a correlation between dopamine D_2/3_R availability and cognitive functioning might be verified in a larger sample. Multiple studies have demonstrated a positive association between striatal dopamine D_2/3_R availability and executive function in healthy individuals [55,56,57,58,59]. Our findings suggest an inverse rather than a positive correlation. This discrepancy might be explained by the inverted U-shaped curve model presented in [60]. According to this model, hypo- and hyperstimulation of the dopamine D_1_ receptor are associated with deteriorated working memory functioning. The inverted U-shaped curve model might also apply to other aspects of dopaminergic functioning, such as dopamine D_2/3_R availability, as well as to other cognitive domains. Moreover, Damsa et al. [61] reported an inverted U-shaped association between learning from negative feedback and striatal dopamine D_2/3_R availability. Additionally, dopamine D_2/3_R availability might only be associated with specific aspects of cognitive functioning, whereas a previous study in healthy individuals found that D_2_ receptor availability in the limbic striatum was related to performance on tests of episodic memory but not to performance on tests of general knowledge or verbal fluency [56]. In addition, the association between dopamine D_2/3_R availability and specific aspects of cognitive functioning might be region-specific, as D_2_ receptor availability in the associative and sensorimotor subdivisions of the striatum of healthy individuals were found to be less correlated to episodic memory but were instead found to be associated with non-episodic tests [56]. Future studies should further investigate the association between cognitive and dopaminergic measures, as well as the potential of dopaminergic drugs to reduce cognitive deficits in 22q11DS.

### 4.4. Strengths and limitations

A major strength of this study is the use of multiple imaging modalities (i.e., 7T MRI and PET) in a sample of adults with 22q11DS who were not psychotic and antipsychotic-free at the time of inclusion. However, some limitations have to be taken into account as well. First, as previously mentioned, the sizes of our MRI and PET samples were small due to the difficulty in recruiting this study population; therefore, this study lacked the power to detect significant associations. Secondly, although the majority of the sample did not use psychotropic medication, two patients with 22q11DS and one healthy control used SSRIs. Because SSRIs indirectly inhibit dopaminergic neurotransmission [62], participants were asked to refrain from this medication on the day of the scanning to limit acute effects on the glutamatergic and dopaminergic systems. Third, we investigated the dopaminergic system during rest and not following pharmacological, behavioral, or cognitive challenges. Therefore, our study does not provide insight into whether other aspects of dopaminergic functioning are altered in 22q11DS. Fourth, the phenotypic expression of 22q11DS is highly heterogeneous and includes congenital heart disease [3]. Consequently, many patients with 22q11DS carry medical implants and were therefore not allowed to participate in the 7T ^1^H-MRS study. In addition, because the majority of 22q11.2DS patients with psychosis use antipsychotic medication and are often not mentally competent to provide informed consent, we did not include these patients in the current study to minimize heterogeneity in the sample. This may have caused a selection bias of relatively healthy patients and made it difficult to generalize the results to the whole 22q11DS population. Finally, although, contrary to 3T, 7T MRI glutamate and glutamine can be reliably distinguished, it does not enable detailed localization of glutamatergic metabolites (e.g. pre- versus postsynaptic and intracellular versus extracellular).

### 4.5. Implications and Suggestions for Future Work

Although we did not find significant associations after Bonferroni correction between dopaminergic, glutamatergic, and cognitive functioning, some associations reached statistical significance. Our findings suggest that the association between ACC volumes and glutamate, glutamine, and Glx concentrations in the ACC are likely differ between individuals with 22q11DS compared to healthy controls. In addition, dopamine D_2/3_ availability seems to be related to cognitive functioning, although the causal relationships between cognitive domains and dopaminergic functioning are yet unknown. Future research with larger samples is needed to further elucidate both of these hypotheses. Furthermore, ACC glutamatergic functioning might not be related to dopamine D_2/3_R availability in 22q11DS but instead be associated with other aspects of dopaminergic functioning, such as striatal dopamine synthesis capacity or dopamine transporter expression. To investigate this hypothesis, additional studies required that make use of other PET and/or SPECT radiotracers (i.e., ^18^F-FDOPA, ^11^C-DTBZ, or ^123^I-FP-CIT) combined with ^1^H-MRS imaging. 

## 5. Conclusions

This exploratory study addresses the relationships between dopaminergic, glutamatergic, and cognitive functioning in individuals with 22q11DS using ^1^H-MRS and ^18^F-fallypride PET. Although our results did not reach statistical significance, the effect sizes warrant future research on this topic. Additional studies with larger samples are needed to further elucidate our findings. 

## Figures and Tables

**Figure 1 genes-13-01672-f001:**
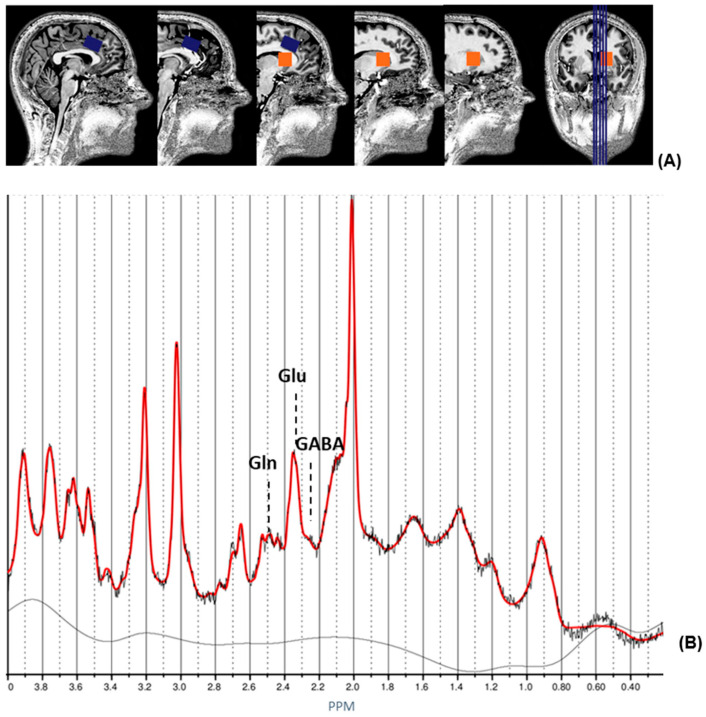
^1^H magnetic resonance spectroscopy (MRS) voxel placement and ^1^H-MRS spectrum. (**A**) Sagittal and coronal views of MRS voxels displayed on a single subject’s T1 structural image. The blue lines in the coronal view (far-right image) indicate the locations of the sagittal views from left to right. The orange box indicates the location of the voxel in the striatum. The blue box indicates the location of the voxel in the ACC. (**B**) Example of an ACC spectrum from a healthy control acquired by LCModel. Reprinted from [21]. Abbreviations: ACC, anterior cingulate cortex; GABA, y-aminobutyric acid; Gln, glutamine; Glu, glutamate; PPM, parts per million.

**Table 1 genes-13-01672-t001:** Sample demographics.

	PET 22q11DS (N = 10) Mean (SD)	MRI 22q11DS (N = 17) Mean (SD)	MRI HC (N = 20) Mean (SD)	Statistic	*p*-Value
Sex (F/M)	5/5	11/6	12/8	0.09	0.77 ^1^
Age, years	37.07 (11.12)	34.17 (11.41)	30.70 (8.20)	145.00	0.46 ^2^
FSIQ	82.60 (12.23)	76.65 (12.32)	120.21 (16.23) ^3^	4.50	**<0.001** ^2^
Smoking in the previous year (yes/no)	0/9 ^3^	2/13 ^3^	2/16 ^3^	NA	1.00 ^4^
Current SSRI use (yes/no)	1/9	2/15	1/19	NA	0.58 ^4^
Time between MRI and PET scan, days	180.10 (349.52)	NA	NA	NA	NA

Abbreviations: F, female; FSIQ, full-scale intelligence quotient; HC, healthy control; M, male; MRI, magnetic resonance imaging; PET, positron emission tomography; SSRI, selective serotonin reuptake inhibitor; 22q11DS, 22q11.2 deletion syndrome. Significant results are bold. ^1^ Differences in sample demographics between MRI 22q11DS and MRI HC samples were assessed using a chi-square test. ^2^ Differences in sample demographics between MRI 22q11DS and MRI HC samples were assessed using a Mann–Whitney U test. ^3^ Data on FSIQ was not available for one HC. Data on smoking status was not available for two patients with 22q11DS and two HCs. ^4^ Differences in sample demographics between MRI 22q11DS and MRI HC samples were assessed using a Fisher’s exact test.

**Table 2 genes-13-01672-t002:** Associations between dopaminergic and glutamatergic functioning in 22q11DS ^1^.

	ACC Glutamate	ACC Glutamine	ACC Glx	Striatum Glutamate	Striatum Glutamine	Striatum Glx
BP_ND_ ^18^F-fallypride ACC	*r* = 0.15 *p* = 0.68	*r* = 0.01 *p* = 0.99	*r* = 0.07 *p* = 0.86	*r* = 0.47 *p* = 0.17	*r* = 0.18 *p* = 0.64	*r* = 0.56 *p* = 0.09
BP_ND_ ^18^F-fallypride CNC (mean)	*r* = −0.33 *p* = 0.35	*r* = −0.27 *p* = 0.45	*r* = −0.46 *p* = 0.19	*r* = 0.21 *p* = 0.56	*r* = 0.27 *p* = 0.49	*r* = 0.17 *p* = 0.65
BP_ND_ ^18^F-fallypride putamen (mean)	*r* = −0.52 *p* = 0.13	*r* = −0.31 *p* = 0.39	*r* = −0.46 *p* = 0.19	*r* = −0.10 *p* = 0.78	*r* = 0.10 *p* = 0.80	*r* = −0.07 *p* = 0.86
BP_ND_ ^18^F-fallypride VST (mean)	*r* = −0.31 *p* = 0.39	*r* = −0.20 *p* = 0.58	*r* = −0.30 *p* = 0.41	*r* = 0.30 *p* = 0.41	*r* = −0.33 *p* = 0.38	*r* = 0.19 *p* = 0.60

Abbreviations: ACC, anterior cingulate cortex; BP_ND_, binding potential; CNC, caudate nucleus; Glx, glutamate plus glutamine; VST, ventral striatum. ^1^ Only correlations with *p* < 0.0083 were deemed statistically significant (0.05/(3 (^1^H-MRS metabolites) × 2 (^1^H-MRS brain regions)); Bonferroni correction).

**Table 3 genes-13-01672-t003:** Associations between glutamatergic functioning and ACC volumes in 22q11DS and healthy controls ^1^.

		Left Rostral ACC Volume	Right Rostral ACC Volume	Left Caudal ACC Volume	Right Caudal ACC Volume
22q11DS	ACC glutamate	*r* = 0.34 *p* = 0.19	*r* = 0.05 *p* = 0.85	*r* = 0.36 *p* = 0.18	*r* = −0.37 *p* = 0.16
	ACC glutamine	*r* = 0.30 *p* = 0.25	*r* = −0.01 *p* = 0.98	*r* = 0.03 *p* = 0.92	*r* = −0.12 *p* = 0.67
	ACC Glx	*r* = 0.01 *p* = 0.98	*r* = −0.30 *p* = 0.26	*r* = 0.14 *p* = 0.59	*r* = −0.43 *p* = 0.09
	Striatum glutamate	*r* = −0.45 *p* = 0.08	*r* = −0.05 *p* = 0.85	*r* = 0.09 *p* = 0.74	*r* = 0.01 *p* = 0.96
	Striatum glutamine	*r* = −0.05 *p* = 0.85	*r* = 0.20 *p* = 0.48	*r* = 0.12 *p* = 0.67	*r* = −0.21 *p* = 0.44
	Striatum Glx	*r* = 0.02 *p* = 0.94	*r* = 0.12 *p* = 0.66	*r* = 0.31 *p* = 0.25	*r* = −0.09 *p* = 0.73
HC	ACC glutamate	*r* = 0.22 *p* = 0.36	*r* = 0.49 ***p* = 0.03**	*r* = −0.14 *p* = 0.54	*r* = 0.12 *p* = 0.61
	ACC glutamine	*r* = 0.09 *p* = 0.71	*r* = −0.15 *p* = 0.54	*r* = 0.51 ***p* = 0.03**	*r* = −0.11 *p* = 0.65
	ACC Glx	*r* = 0.10 *p* = 0.67	*r* = 0.25 *p* = 0.29	*r* = 0.25 *p* = 0.30	*r* = −0.53 ***p* = 0.02**
	Striatum glutamate	*r* = -0.01 *p* = 0.97	*r* = 0.40 *p* = 0.08	*r* = −0.18 *p* = 0.44	*r* = 0.31 *p* = 0.18
	Striatum glutamine	*r* = −0.33 *p* = 0.23	r = -0.31 *p* = 0.26	*r* = −0.07 *p* = 0.80	*r* = −0.37 *p* = 0.18
	Striatum Glx	*r* = −0.16 *p* = 0.49	*r* = 0.14 *p* = 0.54	*r* = −0.39 *p* = 0.09	*r* = 0.15 *p* = 0.53

Significant results before Bonferroni correction are bold. Abbreviations: ACC, anterior cingulate cortex; Glx, glutamate plus glutamine; HC, healthy control; 22q11DS, 22q11.2 deletion syndrome. ^1^ Only correlations with *p* < 0.0125 were deemed statistically significant (0.05/4 (ACC volumes); Bonferroni correction).

**Table 4 genes-13-01672-t004:** Association between cognitive functioning and dopamine D_2/3_ receptor availability in 22q11DS ^1^.

	BP_ND_ ^18^F-Fallypride ACC	BP_ND_ ^18^F-Fallypride CNC (Mean)	BP_ND_ ^18^F-Fallypride Putamen (Mean)	BP_ND_ ^18^F-Fallypride VST (Mean)
Visual memory	*r* = −0.72 ***p* = 0.02**	*r* = 0.21 *p* = 0.56	*r* = 0.36 *p* = 0.31	*r* = −0.21 *p* = 0.56
Verbal memory	*r* = −0.62 *p* > 0.05	*r* = 0.09 *p* = 0.80	*r* = -0.08 *p* = 0.83	*r* = -0.56 *p* = 0.09
Working memory	*r* = −0.63 *p* > 0.05	*r* = −0.03 *p* = 0.93	*r* = 0.24 *p* = 0.50	*r* = −0.26 *p* = 0.47
Attention ^2^	*r* = −0.55 *p* = 0.13	*r* = 0.20 *p* = 0.61	*r* = 0.07 *p* = 0.87	*r* = −0.33 *p* = 0.38
Processing speed	*r* = -0.03 *p* = 0.93	*r* = 0.15 *p* = 0.68	*r* = −0.13 *p* = 0.73	*r* = −0.02 *p* = 0.96
Executive functioning	*r* = −0.74 ***p* = 0.01**	*r* = −0.29 *p* = 0.42	*r* = −0.08 *p* = 0.83	*r* = -0.24 *p* = 0.51
Social cognition	*r* = −0.60 *p* = 0.07	*r* = 0.09 *p* = 0.80	*r* = 0.12 *p* = 0.74	*r* = −0.46 *p* = 0.18
Composite score	r = −0.78 ***p* = 0.01**	*r* = 0.06 *p* = 0.88	*r* = 0.07 *p* = 0.86	*r* = -0.43 *p* = 0.21
FSIQ	*r* = −0.45 *p* = 0.19	*r* = 0.26 *p* = 0.47	*r* = 0.34 *p* = 0.34	*r* = 0.27 *p* = 0.46

Significant results before Bonferroni correction are bold. Abbreviations: ACC, anterior cingulate cortex; BP_ND_, binding potential; CNC, caudate nucleus; FSIQ, full-scale intelligence quotient; VST, ventral striatum. ^1^ Only correlations with *p* < 0.00555 were deemed statistically significant (0.05/9 (seven cognitive domains, composite score, and FSIQ); Bonferroni correction).^2^ One 22q11DS subject was excluded from the analyses that focused on cognitive domain attention due to an extreme value.

## Data Availability

Not applicable.

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
