# Peer review of "The Relationships between Dopaminergic, Glutamatergic, and Cognitive Functioning in 22q11.2 Deletion Syndrome: A Cross-Sectional, Multimodal 1H-MRS and 18F-Fallypride PET Study"

_genes, 2022, doi:10.3390/genes13091672_

Round 1

Reviewer 1 Report

The study is well designed and conducted, and the paper in well written. Although the results are not statistically significant, the results are important to be published.

Minor points:

1 - The authors should clarify and discuss in the paper why only patients with 22q11DS without psychotic symptoms were included. It would be interesting to compare 22q11.2DS patients with and without psychotic symptoms.

2 - The authors should present the differences between glutamatergic functioning and brain volumes between 22q11DS and healthy controls. If it is not possible, the author should clarify that and discuss.

3 - As the cognitive functioning is heterogeneous in 22q11DS the authors could explore the differences in dopamine D2/3 receptor availability between patients with better and worse cognitive functioning.

4 - Some typo errors and minor English mistakes should be revised.

5 - All gene names should be in italics.

6 - The legend of figure 1 should be rewritten with more details.

Reviewer 2 Report

van Hooijdonk and colleagues performed an exploratory case-control study on a small cohort  (17 patients) affected by 22q11.2 deletion syndrome, intending to examine the association between dopaminergic and glutamatergic functioning in this syndrome and their role in the psychotic disorder and cognitive impairment associated to this condition. These hypotheses are supported by the fact that COMT and PRODH enzymes are encoded by haploinsufficient genes included in the deletion, and previous studies, which investigated dopaminergic and glutamatergic functioning in individuals with 22q11DS. Authors apply multiple imaging modalities, i.e. 1H-MRS  and 18F-fallypride PET, to assess glutamate, glutamine, and Glx concentrations in the right striatum and ACC, and dopamine D2/3R availability in the putamen, caudate nucleus, ventral striatum, and ACC, respectively. Neuropsychological evaluation was also performed. Even if no significant associations were found after Bonferroni correction, results suggest: 1) a different association between ACC volumes and glutamate, glutamine, and Glx concentrations in the ACC in patients in respect of controls 2)  a relation between dopamine D2/3 availability and cognitive functioning 3) a possible non-correlation between ACC glutamatergic functioning and dopamine D2/3R availability in 22q11DS patients.

The article is well structured. Methods and results are clearly written. Limitations of the study are adequately reported and discussed. Even if the results are not highly significant, likely due to the small sample size, this work provides important indications for future works aimed at clarifying the neuropsychological aspects of this syndrome.
